# Local Government Approaches to Combating COVID-19 Inequities: A Durham County Department of Public Health Perspective

**DOI:** 10.3390/ijerph18126544

**Published:** 2021-06-18

**Authors:** Kristen Burwell-Naney, Marissa Mortiboy, John-Paul Zitta, Elizabeth Stevens, Kristen Patterson, James Christopher Salter, Michele Easterling, Lindsey Bickers Bock, Hattie Wood, Malkia Rayner, Rodney Jenkins

**Affiliations:** 1Durham County Department of Public Health, Population Health Division, 414 East Main Street, Durham, NC 27701, USA; knaney@dconc.gov; 2Durham County Emergency Medical Services, Office of Emergency Services, 201 East Main Street, Durham, NC 27701, USA; jpszitta@dconc.gov; 3Durham County Department of Public Health, Public Health Clinical Services, 414 East Main Street, Durham, NC 27701, USA; estevens@dconc.gov; 4Durham County Department of Public Health, Public Health Services, 414 East Main Street, Durham, NC 27701, USA; kmpatterson@dconc.gov (K.P.); rejenkins@dconc.gov (R.J.); 5Durham County Department of Public Health, Environmental Health Services, 414 East Main Street, Durham, NC 27701, USA; jsalter@dconc.gov; 6Durham County Department of Public Health, Nutrition Services, 414 East Main Street, Durham, NC 27701, USA; measterling@dconc.gov; 7Durham County Department of Public Health, Health Education Services, 414 East Main Street, Durham, NC 27701, USA; lbickersbock@dconc.gov; 8Durham County Department of Public Health, Nursing and Clinical Services, 414 East Main Street, Durham, NC 27701, USA; hwood@dconc.gov; 9Durham County Department of Public Health, Communicable Disease Programs, 414 East Main Street, Durham, NC 27701, USA; mrayner@dconc.gov

**Keywords:** COVID-19, health inequities, Durham County Department of Public Health (DCoDPH), populations of color, vulnerable communities

## Abstract

When a novel coronavirus disease (COVID-19) made major headlines in 2020, it further exposed an existing public health crisis related to inequities within our communities and health care delivery system. Throughout the COVID-19 pandemic, populations of color had higher infection and mortality rates, and even experienced greater disease severity compared to whites. Populations of color often bear the brunt of COVID-19 and other health inequities, due to the multifaceted relationship between systemic racism and the social determinants of health. As this relationship continues to perpetuate health inequities, the local health department is an agency that has the jurisdiction and responsibility to prevent disease and protect the health of the communities they serve. When equity is integrated into a health department’s operational infrastructure as a disease prevention strategy, it can elevate the agency’s response to public health emergencies. Collecting, reporting, and tracking demographic data that is necessary to identify inequities becomes a priority to facilitate a more robust public health response. The purpose of this paper is to present strategies of how a local health department operationalized equity in various stages of COVID-19 response and apply these methods to future public health emergencies to better serve vulnerable communities.

## 1. Introduction

When a novel coronavirus disease (COVID-19) pandemic took the world by surprise in 2020, it also illuminated existing inequities within our communities and health care delivery systems that disproportionately impact populations of color [1]. These communities are overburdened by poor health outcomes due to the multifaceted relationship between systemic racism and the social determinants of health, and these factors have been the impetus in driving inequalities over the course of the pandemic [2]. According to the Johns Hopkins University and American Community Survey, the COVID-19 infection rate of predominantly African American counties in the United States is 137.5 per 100,000 and the death rate is 6.3 per 100,000 [3]. This represents a six-fold difference in COVID-19 mortality compared to mortality in predominantly white counties. Additional studies corroborate the disparate impacts of COVID-19 on Hispanic communities [4,5], where disease severity is often greater in Hispanic and African American COVID-19 patients compared to white counterparts. Managing COVID-19 inequities requires an understanding of differences in exposure, vulnerability to infection and disease, disease consequences, social consequences, effectiveness of control measures, and adverse consequences of control measures [6]. 

When the World Health Organization (WHO) officially declared COVID-19 a pandemic on 12 March 2020 [7], the first COVID-19 case was also detected in Durham County, North Carolina (NC). The Durham County Department of Public Health (DCoDPH) was tasked with providing COVID-19 surveillance and response to prevent disease transmission and promote health among Durham County residents. DCoDPH is a medium-sized health department located in downtown Durham and offers the following services: (1) Women’s Health Services, (2) Child Health, (3) Communicable Diseases, (4) Dental Clinic, (5) Environmental Health, (6) Health Education, (7) Population Health, (8) Nutrition, (9) STD/HIV Testing, (10) Laboratory, (11) Pharmacy, (12) Vital Records, and (13) Administration. DCoDPH served 10,884 individuals during the fiscal year 2019–2020 and normally operates on a budget of roughly 26 million dollars per year. DCoDPH has approximately 200 employees who are dedicated to executing a vision of working with the community to prevent disease, promote health, and protect the environment. The organizational leadership structure of DCoDPH consists of the health director, two deputy health directors, and nine division directors that oversee all programs and services. The Durham County Board of Health is responsible for creating policies and mandates that allow DCoDPH to operate at its optimal potential. 

DCoDPH has long operationalized a stance on racism as a public health issue. In combating the COVID-19 pandemic, the health department has acknowledged the influence of racism on health care delivery and ways that the social determinants of health may cause certain communities to be less resilient to such a complex disease [2]. As an agency working with community partner collaborations, DCoDPH has viewed the pandemic with an equity lens seeking to protect the health of the most vulnerable populations in the Durham community. When equity is integrated into a health department’s operational infrastructure, it can elevate the agency’s response to public health emergencies by prioritizing the collection, reporting, and tracking of demographic data that is necessary to identify and respond to inequities [8]. Despite experiencing challenges while navigating the pandemic, this process has cultivated an innovative preparedness system that will allow DCoDPH to more confidently respond to any pandemic, disaster, or other public health crises that may arise in the future. The purpose of this paper is to present strategies of how a local health department can optimize equity in various stages of COVID-19 response and apply these methods to future public health emergencies to better serve vulnerable communities. 

## 2. Materials and Methods

### 2.1. Study Area

Durham County is centrally located in the northern region of North Carolina and is considered the sixth most populous county in the state. The demographics of Durham County residents have shifted significantly over the last two decades. Since 2000, the population has grown over 64% to 311,848 residents in 2019 [9]. Estimates for 2019 show that non-Hispanic African Americans and non-Hispanic whites comprise most of Durham’s population, 36.5 and 51.9%, respectively. Native Americans, Asians, and other ethnicities account for the remaining 11.6%, while Hispanics make up an estimated 13.5% of the county population [9]. In 2019, the proportion of residents who spoke a language other than English at home was 18.6%. 

### 2.2. COVID-19 Data Infrastructure and Case Information

The Durham County COVID-19 line listing is an Excel spreadsheet that contains data about each positive case. The line listing was developed to serve as a central database for COVID-19 disease surveillance and case management in response to the COVID-19 index case identified on 12 March 2020. It is used as a data hub to perform all COVID-19 data analyses and data feed for our public facing dashboards. The line listing was created in Excel Online to provide easy access to those who needed use of the file to allow quick editing as needed. Over time, the line listing became a large very complex Excel sheet, containing over 25,000 records and 132 columns. To ensure that the case volume was entered and managed in a timely manner, multiple personnel were granted editing privileges such that approximately seven people were working on the document at any given time. Additional personnel who needed access to the line listing to perform job specific duties were granted view only access to reduce the likelihood of error when managing a large database. 

The variables selected for data collection were centered around reporting requirements developed by the North Carolina Department of Health and Human Services (DHHS), the state’s public health department. These variables included a unique COVID-19 ID number, contact information (e.g., phone number and email address), patient name, date of birth, and physical address. Demographic data recorded included race, ethnicity, age, sex, and occupation or student status. Race and ethnicity data were self-reported by positive cases. The validity of race and ethnicity self-reporting was verified by sources providing positive test results and when DCoDPH surveillance staff contacted the positive case. DCoDPH utilized the standard practice for operationalizing race and ethnicity; we communicated with the State of North Carolina Division of Public Health, verified and standardized our methodologies for all case data for reporting purposes. The state of North Carolina also provides our data from a shared database, and we follow their naming conventions on all race and ethnicity data.

While never mandated by the state, the health department began collecting occupation and employment information as well as data on reinfections. Over 17,000 unique data entries for employment and occupation were collected to assist with tracking epidemiological clusters and identifying potential trends in cases. Additional data columns in the line listing included a notes section that contained brief information about the patient, such as where an individual may have contracted COVID-19, family members who may also be positive, date of reinfection, travel, and other relevant data. The line listing contained specific columns for use by the outbreak team with names of schools, workplaces, residential communities, nursing home facilities, or other significant points of transmission throughout the community. These columns were added to assist the outbreak team in rapid and efficient detection of disease clusters, to facilitate an appropriate public health response. COVID-19 case outcome data was provided through the line listing. DCoDPH staff disaggregated the data for various ethnic or racial groups to gather a deeper understanding of trends so biases were not used in decision-making or when developing reports. The line listing in Excel was connected to Power Bi, another Microsoft tool used for data collection, analysis, and visualization. Power Bi was the primary analysis tool used in daily surveillance efforts, along with Python and its various libraries and SAS 9.4. Power Bi allowed the team to connect directly to the line listing and perform analyses by quickly summarizing columns and values and creating new measures and columns from existing data. The line listing data was used to compute a 7 Day Moving Average (7DMA) of active cases, total hospitalizations, case distribution by zip code, outbreak information, and breakdown by month or day of the pediatric population. Power Bi was primarily used to view and communicate race and ethnicity data and to identify demographic trends. This information led to the development of actionable solutions through social media messaging and directed outreach programs when applicable. Through Power Bi, the ability to cross reference data from emergency services and the line listing served as an alert system to notify emergency service partners when they may encounter a COVID-19 positive patient. Use of Power Bi contributed to a more efficient management response to encounters with COVID-19-positive patients. Moreover, the health department provided the local 911 Call Center with certain address data points so when they responded to a call for a COVID-19 positive patient at a nursing care facility or community home, they would be notified to wear proper personal protective equipment (PPE). The dispatcher would only communicate to the Emergency Services provider to wear proper PPE as to not identify infected individuals. 

The Power Bi demographic data pages were used to inform weekly trends in COVID-19 case data, as well as detect missing data as a validation measure. As the number of COVID-19 cases increased, the volume of missing demographic, employment, and contact information data increased as well. Power Bi was used to locate missing data on a row-by-row level so that case investigators could reach back out to a specific case and collect the necessary information required to achieve a complete dataset. Other data validation methods applied in Power Bi included the conversion of data column types to numbers if they were entered as text in the line listing and the identification of misspellings in Power Bi that could be addressed and corrected in the line listing Excel sheet. Columns containing dates were validated and rules were applied that only allowed dates to be entered in the corresponding fields. Type casting was applied to all columns in Power Bi and a Primary Key, known as Event ID in the line listing, was used to verify unique rows. In addition, data from the line listing was validated against the North Carolina Electronic Disease Surveillance System (NCEDSS) for COVID-19 to ensure that all Durham County cases documented in NCEDSS were captured on the line listing. 

All COVID-19 case data were managed in compliance with Health Insurance Portability and Accountability (HIPAA) guidelines provided by U.S. DHHS to protect patient confidentiality and protected health information (PHI). All tables that contained any race or ethnicity data were independent of the row level data and did not possess any identifying columns, such as name (first/last), date of birth, and address data. U.S. regulations allow for the use and release of race and ethnicity data for research and educational purposes. The COVID-19 data collected and analyzed falls within those guidelines and was used to make informed detailed decisions. The race and ethnicity information included in the data set allowed DCoDPH to measure trends about populations most impacted in real time, such as increased cases among the Hispanic population, respond quickly, develop tailored responses, and share information with community partners.

### 2.3. COVID-19 Staffing and Operations

The Durham County Emergency Operations Center (EOC) was activated in March 2020 in response to the first cases of COVID-19 within its jurisdiction. DCoDPH leadership was integral in planning the pandemic response. Staff from each division were pulled from their regular job duties to assist with COVID-19 response activities, such as developing COVID-19 screening procedures and tools for individuals entering buildings, answering a telephone hotline set up for the public, conducting case investigation and contract tracing, and collecting data. Health department clinics were suspended or scaled back to a limited schedule so staff could focus on COVID-19 response activities. Once the scope of the pandemic became clearer, DCoDPH set up teams to focus on the needs of vulnerable populations. 

The Homeless Task Force worked with local shelters to prevent outbreaks by developing social distancing plans, screening tools, and strategies to meet food and housing needs for the local homeless population. The Community Task Force focused more on the Hispanic population due to an early increase in case rates within this population. This task force conducted Spanish language media outreach and worked with Spanish-speaking staff and contractors to address the population’s social needs and build trust. The Food Security Task Force convened anti-hunger advocates and food pantry staff and volunteers along with county staff and elected officials to monitor and combat the dramatic rise in food insecurity. They worked to ensure that anti-hunger agencies had the resources to continue providing services safely while meeting the higher demand. 

Two strike teams were formed to respond to outbreaks of COVID-19 cases in the community. The Long-Term Care Facility Strike Team communicated with all residential living facilities, including homeless shelters and detention facilities, sharing guidance and performing site visits to provide recommendations based on Centers for Disease Control and Prevention (CDC) COVID-19 guidance. The Clusters Strike Team concentrated on locations with a higher prevalence of cases, such as workplaces, churches, schools, and daycares. Interventions and activities by the various teams focused on vulnerable populations were grounded in real-time data and trends with regular reference to the aforementioned data being collected by the health department as a whole. Strike teams were necessary to narrow the focus and provide a more targeted response for vulnerable populations most impacted by COVID-19, such as those experiencing homelessness, job loss, or language barriers. Trends within the strike teams were also noted, to focus attention on higher rates of infection among racial or ethnic groups within specific clusters of cases, and to inform mitigation strategies within these settings.

### 2.4. COVID-19 Testing

The DCoDPH COVID-19 testing strategy aimed to make testing free, easy, and accessible to all members of the community. The health department used spatial statistics based upon zip codes with the highest rates of infection. DCoDPH was able to work with testing partners such as North Carolina Department of Health and Human Services (NC DHHS), Optum Serve, CVS, and Walgreens to deploy mass testing in targeted geographic locations to allow for the identification of pockets of infection. The goal was to isolate and quarantine infected populations as quickly as possible. 

The health department created and utilized the Durham County Coronavirus Data Hub website as a public-facing resource for factual, evidence-based information to drive decision making. Through strengthened partnerships with community agencies, testing sites with accessible hours were spread strategically throughout the county and located on public transportation routes. Partner locations positioned the health department for outreach to disproportionately impacted African American and Hispanic populations. 

DCoDPH and the Duke Division of Community Health set up a free testing site in summer 2020 at the Holton Career and Resource Center in East Durham to target close contacts to positive cases. This was in response to peaking cases among the Hispanic community, which represents 14% of the county population, but accounted for 71, 76, and 53% of COVID-19 cases in May, June, and July 2020, respectively. DCoDPH provided scheduling for the site, and the Duke Health team provided testing three times per week during evening hours and Saturday mornings. The Holton Career and Resource Center site operated from 11 July 2020–25 February 2021 and served as a drive-thru location that was convenient for many Durham residents. 

Testing communication strategies included advertising in Spanish through multiple media channels, publishing testing locations on the DCoDPH website, and providing the information to the COVID hotline and case investigators to share with callers. Contact tracers reaching out to close contacts automatically signed willing individuals up for COVID-19 testing at the Holton Career and Resource Center site. 

Barriers such as transportation, accessibility, and language made it more challenging for certain segments of the population to obtain COVID-19 tests. Efforts to place testing locations in convenient and familiar community locations and concerted attempts to distribute information in languages other than English assisted with increasing testing levels among historically marginalized populations.

The DCoDPH Laboratory gained the ability to provide gold-standard polymerase chain reaction (PCR) COVID-19 testing in house, which was offered specifically to the detainee population at the Durham County Detention Facility. This guaranteed a fast turnaround time for testing of these vulnerable individuals, to help identify and control any cases within this high-risk setting. 

### 2.5. COVID-19 Resource Allocation

Throughout 2020, the DCoDPH provided flyers in English and Spanish with accurate COVID-19 information to community partners such as Durham Housing Authority, Durham Public Schools, and local non-profits such as End Hunger Durham to distribute to community members. The DCoDPH Health Education and Community Transformation Division facilitated outreach events in apartment complexes with clusters or high number of cases. Additional outreach was done at Spanish grocery stores shortly before Thanksgiving at locations within city limits. The Environmental Health division provided additional resources and guidance to long-term care facilities such as nursing homes, assisted living centers, and other congregate living establishments. Resources provided to these facilities included the following: PPE, diagrams for donning and doffing, required and recommended signage, infection control tools, and extensive information on COVID-19 testing available in the Durham community. On numerous occasions, antigen and PCR test kits were provided to these facilities so that “in-house” samples could be collected before shipping to commercial laboratories or the NC State Lab of Public Health (SLPH) testing facilities. 

The health department provided food when this need was identified during COVID-19 surveillance calls. To qualify for food delivery, a household member had to be COVID-19 positive and in a quarantine or isolation status, followed by the health department for daily surveillance, and lacking the social or financial support to have food delivered in another way. This reduced food insecurity, allowed under-resourced individuals to maintain health as much as possible, and likely reduced additional disease transmission by enabling individuals to adhere to quarantine guidelines. The health department also provided thermometers, cleaning materials, face masks, fever reducing medicine (e.g., Tylenol), infant formula, and diapers if needed. Resources were provided to those who self-reported a need or had inability to pay for supplies or food during the quarantine period. Food and supplies were delivered seven days a week, including holidays, and were delivered within 24 h of the referral. DCoDPH ceased delivering food in October 2020 and began referring households to similar services being offered through the Duke COVID-19 Social Support Program. 

### 2.6. Partnership Involvement

DCoDPH worked with a variety of partners across multiple sectors to ensure COVID-19 monitoring, surveillance, testing, and resource needs were met throughout the pandemic. Table 1 provides a list of collaborative DCoDPH partners throughout the COVID-19 pandemic. The list identifies the sectors needed to adequately respond to COVID-19 needs among Durham County residents. Given the physical distancing required during the COVID-19 pandemic, DCoDPH used multiple community engagement strategies to work with partners to provide outreach, education, and services related to COVID-19, including town halls, Facebook Live sessions, Zoom meetings, feedback from DCoDPH data dashboards, press conferences, briefings to elected officials, radio interviews, social media, email communication, and weekly video messages from the health director to the Durham community. Communication strategies were adjusted throughout the COVID-19 response based on feedback from trusted community partners. Lack of coordination is a perpetual challenge when addressing health issues in the community. Therefore, intentional efforts were made to prevent this during regular tactics meetings, by facilitating direct information sharing among community partners working directly with vulnerable members of the community.

Health department staff regularly attended meetings with community collaboratives and shared data. Staff returned with input from the meetings to health department leadership to inform decisions. DCoDPH also worked with community partners to solicit feedback for the Coronavirus Data Hub website and weekly infographics and reflections. In many cases, community partners provided DCoDPH with a more detailed view of the needs and challenges experienced by vulnerable members of the community. DCoDPH was a part of the Recovery and Renewal Task Force, a coalition of influential stakeholders convened by the Mayor of Durham and the Chair of the Durham County Board of County Commissioners. These new and existing relationships with community partners expanded reach to historically marginalized populations to provide accurate information, distribute outreach materials, and support community needs. 

### 2.7. COVID-19 Data Dissemination

Multiple dissemination strategies and mediums were utilized during the pandemic to keep community members, partners, and government officials updated on the most recent COVID-19 data and trends. The Durham County Data Hub was hosted by ArcGIS and was established on 4 April 2020 as DCoDPH’s primary public-facing dashboard. Prior to releasing the dashboard, DCoDPH spoke with community organizations representing the Hispanic and African American populations to identify the best methods to display the data and add context to avoid stigmatizing this population. DCoDPH representatives also attended community meetings and met with partners to gain input before the dashboard was published.

The Data Hub was updated once daily to provide real-time case data related to the following sections: (1) case data overview, (2) weekly data reflections and infographic, (3) percent positivity graph, (4) testing facility or vaccine provider site locator, (5) Durham County case count by zip code map, (6) Durham County case demographic statistics, (7) local GIS resources, (8) social media and official updates, (9) resources, (10) symptoms, and (11) prevention information. The case data overview sections contained data on total confirmed cases, active cases, demographic information (i.e., age, gender, race, ethnicity), occupation data, deaths, and outbreaks in congregate living settings, schools, and childcare centers. 

Weekly data reflection reports and infographics were posted on the Data Hub every Friday with data from the previous week. Data was imported into RStudio from the line listing as a csv. file with columns for create date, age, gender, race, ethnicity, employer, and occupation. North American Industry Classification System (NAICS) codes were matched with employer information, and all duplicate records were removed. Graphs were created to represent the percent of COVID-19 cases by race, ethnicity, industry, and workers in each industry by ethnicity, race, and gender. Percentages of COVID-19 cases were calculated by race and ethnicity and compared to the demographic composition of those subgroups for Durham County. Sections of the report were developed specifically to help readers understand the importance of stratifying data related to race and ethnicity, how to interpret race and ethnicity data, and whether the percentage of COVID-19 cases were representative of the respective population in Durham County. The percentage of missing data were also calculated for race and ethnicity as well as disease rates by zip code. An infographic was developed from the data presented in the weekly data reflections report to highlight COVID-19 data trends and data by race, ethnicity, and occupation. 

Zoom calls were held with community partners from LATIN-19, Partnership for a Healthy Durham, Isla-NC, the African American COVID-19 Task Force, and many others. Calls conducted with community partners often involved a representative from the health department (e.g., Health Director, Deputy Health Director, Population Health Division Director, or a Public Health Education Specialist) giving a presentation on COVID-19 updates, performing a tutorial of the Data Hub dashboard, including specific demonstrations of how to display data sorted by race and ethnicity, and leading a discussion eliciting feedback on the utility of the resources available and what improvements were needed to meet the needs of the community. One in-person meeting was held with Dr. Mandy Cohen, Secretary for the NC Department of Health and Human Services. 

DCoDPH’s Environmental Health Division Director worked closely with the Mayor’s office and other organizations to organize a team of Health Ambassadors from the University of North Carolina for in person dissemination of COVID-19 information. The team consisted of approximately 30 field staff, more than half of whom were fluent in Spanish. This was an effort to more clearly communicate the need to adhere to mask wearing, social distancing, quarantine, isolation, testing, and other infection control actions needed to slow the spread of COVID-19 in the Hispanic community. 

In addition, regular mass emails were sent from the Environmental Health office and the Public Education Unit in English and Spanish regarding important information and resources relative to COVID-19. This included guidance on PPE requirements in nursing homes, information on the Greenlight Durham program, which helps businesses access testing, location of COVID-19 testing sites, and other pertinent materials. Environmental Health e-mail blasts were sent to roughly 3000 recipients at a time. Public Education e-mail digests, summarizing CDC guidance, state executive orders, local services, resources, and data were distributed five days per week from March–August 2020, and then three times per week from September 2020–May 2021.

Environmental Health also worked with the City Attorney’s office to help interpret state laws and rules that applied to private clubs, bars, and restaurants, in response to the massive number of complaints regarding non-compliance with COVID-19 executive order requirements. Multiple communication channels such as word of mouth from trusted sources and electronic and traditional media were needed to accommodate different populations and reinforce messaging. Vulnerable populations are often not reached by general messaging; therefore, targeted and culturally appropriate communication efforts were essential to effectively reach different audiences across the county. 

## 3. Results

### 3.1. COVID-19 Data Infrastructure and Case Information

The data collection infrastructure and methodology allowed DCoDPH to comprehensively capture the demographic data points necessary to address inequities in COVID-19 transmission throughout the community. Figure 1 shows the total number of COVID-19 confirmed cases by race and ethnicity. The information in this figure reflects data captured from March 2020 through May 2021. The highest percentage of total COVID-19 cases have been among African-American residents (38%) who make up 37% of the Durham County population, followed by Hispanic residents (32%) at 14% of the population and whites (28%), who comprise 54% of Durham County residents. 

There were very few data points missing in the DCoDPH line listing. Figure 2 shows missing demographic information by ethnicity, race, age, and gender by quarter. The greatest percentage of missing data for ethnicity (6.3%) and race (4.9%) were reported in quarter 1 of 2021, which included January–March 2021 data. The highest missing data for age occurred during quarter 4 of 2020 (0.9%), and gender was most incomplete during quarter 2 of 2021 (0.5%). Quarter 3 of 2020 then had the least amount of data missing for all categories combined (1.4%). 

### 3.2. COVID-19 Staffing and Operations

From March to December 2020, approximately 150 DCoDPH staff and contractors were involved with COVID-19 case investigation and contact tracing efforts. DCoDPH hired additional Hispanic contractors from Community Care of North Carolina (CCNC) and COVID-19 Community Team Outreach (CCTO) to assist with case interviews and contact training. Staff were hired who were bilingual and bicultural: speaking Spanish and possessing a familiarity with the culture, a sense of trust, and relationships with members of the Hispanic community. This proved essential in order to effectively complete case interviews and collect information about contacts. A total of 42 CCNC staff (19 bilingual), 7 U.S. Public Health Service members (1 bilingual), 9 contract agency nurses, and 10 Physician’s Assistant program students from Duke University were brought onboard to assist the health department with COVID-19 surveillance and response. The health department directly funded the employment of the contract agency nurses, while other personnel were funded through outside sources. The increase in workforce allowed the health department to maintain robust surveillance operations on a seven day per week schedule from March to September 2020. Normal operations would have consisted of 127 days during that time period; adjusting the schedule to include weekend days resulted in 55 additional workdays directed at COVID-19 mitigation activities. As case counts allowed, DCoDPH operated six days per week, excluding Sundays and holidays between September 2020 to February 2021. Working a six day per week schedule added 25 days beyond normal hours of operation (118 days) over the six-month period. Beginning in March 2021, DCoDPH resumed normal hours of operation at five days per week. 

Inspections, site visits, and some outreach activities for COVID-19 were conducted by Environmental Health staff. A total of 3235 establishments were visited or inspected from March 2020 through March 2021, including restaurants, childcare centers, tattoo artist shops, residential care facilities, nursing homes, and hospitals (Table 2). Most visits occurred at establishments located in zip code 27701, which has the highest percent poverty in the Durham County (34.8%) [10] and was often one of the top five zip codes with the highest COVID-19 case rate throughout the pandemic. Zip code 27707 has the third highest percent poverty in Durham (17.6%) [10] and was also among the top five zip codes with the highest COVID-19 case rates. Since the percent of COVID-19 cases in the Hispanic community significantly increased from May–June 2020, establishments in these communities were targeted for inspections and outreach visits. Approximately 158 establishments were targeted, including 47 restaurants, 11 food stands, 64 mobile food units, 25 push carts, and 11 meat markets. 

### 3.3. COVID-19 Testing

The Holton Career and Resource Center was a great asset to communities of color in Durham County due to proximity of the facility to these populations and the convenience of a drive-thru site and nearby access to public transportation. The location was a trusted community center, which combined with evening and weekend operating hours helped reduce barriers to accessing testing. This partnership allowed DCoDPH to reduce infections dramatically in Durham through early identification among close contacts. Contact tracers calling close contacts of positive cases automatically offered this resource and signed up willing individuals for COVID testing. Based on the nature of COVID-19 spread, it was presumed likely that many close contacts of positive cases would end up becoming positive themselves. By quickly offering a free and accessible testing option for early identification of these additional COVID infections among the higher-risk close contacts, this partnership allowed the health department to significantly control the spread of COVID-19 within this vulnerable population. The overall positivity rate at the Holton Career and Resource Center was 21.19%, illustrating the high prevalence of infections among the close contact population that were detected due to this accessible testing option.

In the DCoDPH line listing, documentation was included for the people who tested positive at the Holton site, and this was indicated for 86% of the cases tested at Holton. Of the 759 cases the health department identified in the line listing with Holton marked as the testing site, almost 50% of the cases were Hispanic and approximately 27% of cases were African American (Table 3). 

Optum Serve opened several community-based COVID-19 testing sites in October 2020 at strategic locations around the community to increase testing available to historically marginalized groups. Demographic data for those tested at these sites is displayed through 9 April 2021 (Table 4). Approximately 55% of the people being tested at an Optum Serve testing site were identified as individuals from populations of color. Roughly 20% of persons being testing at these sites identified as Hispanic. This ability to capture a high percentage of populations of color was due to the strategic placement of sites within the County. 

### 3.4. COVID-19 Resource Allocation

From the beginning of the pandemic through December 2020, 7417 COVID-19 hotline calls were received from community members requesting information ranging from COVID-19 transmission and testing site locations, food resources, medical questions, and COVID-19 guidance. 

A total of 1049 individuals were provided food while in isolation with COVID-19 from May to October 2020, which equated to 266 households served and $32,991 spent on food during that period. Food included basic provisions such as seven days of food and supplies including fruits and vegetables, meat, fish, oil, eggs, milk, cheese, bread, rice, beans, pasta, soup, toilet paper, bleach, and acetaminophen. Early in the pandemic, DCoDPH staff spoke with impacted families and learned exactly what food items they wanted us to purchase. 

As cases increased, DCoDPH had to change its approach. The health department reviewed common items people were requesting and developed a standard list that accommodated those in different racial, ethnic and cultural groups, and of varying family sizes. For example, the Hispanic families typically preferred dry small red beans or black beans while the African American and white families preferred canned pinto or black, so we provided a combination of the two. The majority of the families wanted whole milk, so that is what was provided. Most families at the time of food delivery were Hispanic, and this ensured their food preferences were met. 

Between July and October 2020, 2750 outreach bags were distributed door-to-door in eight communities with a high number of new COVID-19 cases. Outreach bags included masks, hand sanitizer, 3 Ws magnets (the Wear a mask, Wait six feet apart, and Wash your hands NC DHHS message), coloring books, educational materials, and community resource sheets with information about food and housing. DCoDPH provided English and Spanish education and information via an email list serve directed toward the public, healthcare providers, businesses, faith communities, and community agencies. Through the end of 2020, the health department had sent 676 electronic digests to a total of 325,121 contacts. Almost 2350 different information sources were distributed, including medical guidance, information about local and state stay-at-home orders, and links to Durham resources. Content included in the Spanish digests was curated to meet the needs of Hispanic individuals, while also sharing resources for organizations serving Spanish speakers.

### 3.5. COVID-19 Partnership Involvement

Involving community partners at various points throughout the pandemic was critical to informing how DCoDPH engaged with vulnerable populations within the county. Ten meetings involving Health Education and Community Transformation staff, Bull City United outreach workers, and Curamericas volunteers were held to plan outreach activities and grocery store outreach. A team of healthcare workers at Duke Health began meeting in March 2020 to address concerns about how the pandemic was affecting the Hispanic community. The Latinx Advocacy Team and Interdisciplinary Network for COVID-19 (LATIN-19) emerged as a formalized working group, and health department staff began participating in those meetings in April 2020. The African American COVID Task Force has been meeting weekly since June 2020 and health department staff have participated in those meetings since its inception. 

The need for food and housing resources was illuminated through partner and community feedback. The inclusion of child-focused materials in the outreach bags delivered to targeted apartment complexes was the result of partner feedback and referencing COVID-19 case data. Curamericas conducted formative research with community members in populations of color to find out directly from them which types of masks they found most useful. This information was used to make decisions on the procurement of masks. In addition, community members shared their experiences in reaching out to community organizations for COVID-19-related needs, such as which resources were or were not helpful. Individuals informed DCoDPH about the type of resources that should be shared with those testing positive for the disease. 

Through consistent engagement with the African American COVID Task Force, DCoDPH was able to determine that the weekly data reflection report was too complex for the intended audience. Based on community feedback, DCoDPH developed a weekly infographic to accompany the weekly data reflection report. This weekly infographic comprised the same information but was designed using a simpler format that was easier to understand. The weekly infographic was further updated, using feedback from LATIN-19, to document the percentage of cases in the Hispanic community and calculate the percentage of this population who were represented in occupational cases. Ethnicity data was also added to the weekly infographic, along with the percentage of the Hispanic population represented in the top ten most impacted occupations. 

The Food Security Task Force worked through and supported partner agencies to increase food security. The County funded EAT NC to provide meals to 400 families of children and seniors weekly. The meals not only helped families remain food secure, they also kept multiple women and African American owned restaurants in business. The County funded End Hunger Durham to provide meals for 900 high-risk, low income seniors; El Centro Hispano to provide food aid to 300 Hispanic households; and Meals on Wheels to provide meals to an additional 100 high-risk, low income seniors. Furthermore, the County provided nonprofit grants to five nonprofit anti-hunger organizations so they could meet the higher demand for food related to the COVID-19 pandemic. The Task Force partnered with United Way to set up a fund where Durham residents and companies could donate money to be allocated to anti-hunger agencies applying for funding. The Task Force, DCoDPH, and partners worked collectively to create a searchable food resource map where people in need of food could find free resources. 

### 3.6. COVID-19 Data Dissemination

DCoDPH saw an exponential increase in the number of people utilizing existing health department social media and web resources as a reliable source of COVID-19 information. These channels provided an easy avenue to disseminate data and other information that was rapidly changing. Given early trends in case rates within the Hispanic community, there was an intentional effort to ensure that all information shared through the Public Education Unit and social media was provided in both English and Spanish. The accessibility of this information was received positively. The number of Facebook followers increased from 1332 the week of 2–8 February 2020 to 7132 the week of 21–27 March 2021, which represents a 435% total increase in followers. The number of page views started at 74 during the week of 2–8 February 2020 and surged to a median of 349 views. The highest total number of views occurred on 10–16 January 2021 with 2304 views. Growth was also seen on Twitter, where there were 1826 followers on 1 February 2020 and 3432 ending on 22 March 2021, representing an 88% increase in new follows. The median number of link clicks was 90 and the highest number occurred during the week of 7–13 March 2021 (1100). There were 1697 new subscribers to the daily email digest sent by the Health Education & Community Transformation Division. 

Since the official launch of the Durham County Data Hub on 20 April 2020 through 20 April 2021, there have been 254,182 total visits to the webpage. There were a total of 161,385 unique page views where the average time spent on the site was 1.10 min. Most of the Data Hub users originated in Durham (80%) and there have been approximately 4100 downloads of materials from the site. 

## 4. Discussion

### 4.1. COVID-19 Challenges

Responding to a once-in-a-lifetime public health event of high complexity, large scope, and long duration has included many challenges for public health agencies. Difficulties experienced were not unique to DCoDPH, but can be examined to help other health departments learn, plan, and adopt policies that best serve the public. Staff burnout and turnover was significant. Between March and September 2020, the health department conducted COVID-19 surveillance seven days each week and some staff were working long shifts. There were also inequities discovered in staff scheduling for COVID-19 response, as well as who was eligible to receive overtime and hazard pay. Spanish-speaking staff were in high demand for many COVID-19 response tasks. The addition of contractors helped relieve some of the burden, but did not resolve the need for health department staff involvement. For more than a year, staff were challenged with completing normal job duties in addition to COVID-19 responsibilities.

Many DCoDPH essential public health functions could not be put on hold due to the pandemic. In many cases, client needs were increased, related to challenges associated with COVID-19. Multiple programs were stressed by the inability to adequately serve clients due to safety restrictions on outreach. Many staff experienced challenges with childcare and remote schooling. Several health department staff retired, resigned, or filed for ADA accommodations to work off site during this time related to the stress and pressure of the COVID-19 pandemic. There have been concerns expressed related to the workload staff were asked to take on and the risk for exposure to COVID-19 at work.
Access to testing (transportation, hours, etc.) will never be perfect. There remain individuals who cannot access a testing site. Accessing testing is problematic, especially if an individual is quarantined as a close contact to a positive case. There are also areas within the county where public transportation is limited or nonexistent.The translation of data and complex information into third grade or less literacy levels has proven difficult.Electronic distribution of information created a barrier for certain populations who do not have access to the internet or knowledge of how to use electronic devices. On a related note, the reliance on virtual communication created a barrier for individuals without access to the internet or digital devices being unable to participate in virtual meetings. Individuals participating by phone in virtual meetings were at a disadvantage since they could not participate equally.The nature of the pandemic limited the ability to conduct face-to-face outreach or medical assistance due to COVID-19 restrictions and safety measures.Keeping data clean and providing innovative insight to address data issues was difficult. Minor issues revolving around the line listing data not updating, public dashboards breaking, verifying, and correcting data points often took hours to resolve.Working within the confines of County Information Technology (IT) was sometimes limiting. Using other data platforms could have made data collection and integration more efficient; however, County IT policies and protocols may prevent the use of available platforms.

### 4.2. COVID-19 Lessons Learned

The health department is the cornerstone of many communities due to its role of protecting the health of the public, promoting health equity, and providing services that reduce disease risk. As these agencies encounter various disasters, crises, and pandemics, it is important that health departments learn from past experiences and adapt to the rising challenges of the communities they serve. Throughout the pandemic, DCoDPH has learned several lessons that can be scaled to strengthen the pandemic response of any health department.
View the population you are trying to serve through an equity lens. Decide who should be in leadership positions for pandemic response so that the team reflects the demographic of the population being served. Ensure that staff with the experience, knowledge, and lived experience of the vulnerable communities are given a voice at the table for decision making.Mandate the collection of race, ethnicity, and other important demographic characteristics as a routine public health practice. While there is no formal directive on collecting this information, it has been recommended in other studies and ultimately became the driving force in the ability of DCoDPH to effectively address inequities in the community [11].Listen to staff, both the good and bad. Staff are often on the front lines doing the work and offer a certain expertise that may not be apparent to those in leadership positions with broader views of the organization. Frontline staff may have ideas and opinions about the work to improve efficiency, impact, and equity. This also allows staff to feel heard and contribute to operations.Listen to community organizations regarding their needs. Engage with and build authentic relationships with community members. Do not try to tell them what they need or solve problems for them. Use available resources from the health department to facilitate problem-solving within these communities.Create an environment that fosters innovation through teamwork and avoid working in silos. Responding to a pandemic may require innovative solutions that are often achieved when people contribute their individual expertise and experience to the discussion.Determine appropriate ways to share resources, data, and powerful information with community members and organizations since they are considered the experts of their respective communities. Work with community partners regarding how to display data and what type of information to include.Utilize data tools and technology to their full potential, while allowing for innovation and collaboration across multiple documents. SharePoint is an example of a very powerful tool that allows this type of collaboration. Other data tools such as Microsoft Azure lends itself to connect disparate data and provide a high quality and innovative dashboard for public or internal use.Constantly reassess and adjust operations as needed based on new information or data. This iterative process positions the health department to be prepared to address any issue that may arise in a timely manner. It also fosters an environment that promotes the collection of a complete dataset for evaluating inequities in disease transmission.Provide culturally appropriate and sensitive resources, such as food and educational materials, to the community. Develop materials that are appropriate for different racial/ethnic groups and rely on community partners for review and feedback. Also, engage community partners on the types of food that are culturally appropriate for their communities.Mental health is important. Determine effective strategies to keep staff mentally healthy. Be aware and sensitive to what new and current staff have been dealing with throughout the crisis and respond accordingly. Assign someone to keep a pulse on the morale, health, and well-being of staff. Send out mental health resources on a regular basis and provide reminders of resources at staff meetings.Designate times to celebrate successes and congratulate staff on their performance. Assign times during celebrations to further discuss internal organizational inequities and how staff are doing.Mandate time off for staff. Each staff member should have at least one person who can fill in and perform the equivalent duties so staff can take a true break. Establish scheduling rules that provide consecutive days off to allow for personal tasks, rest, and decompression.

## 5. Conclusions

DCoDPH’s focus on equity in COVID-19 response efforts has strengthened the capacity to actively respond to future pandemics and tailor that response to meet the needs of the most vulnerable populations in the Durham community. Promoting health equity through actionable solutions is a central tenant of DCoDPH’s core values and policies. Over the course of the COVID-19 pandemic, DCoDPH has been able to operationalize this core value by being transparent with the data and offering real-time information to the community. DCoDPH wanted residents to be aware of their disease risk and understand how COVID-19 may be impacting different communities across Durham County’s population. In addition, the health department wanted to provide reliable data that could be used by elected officials, various organizations, and the Emergency Operations Center to inform decisions regarding populations disproportionately impacted by COVID-19. DCoDPH’s commitment to maintaining this level of transparency during the pandemic to mitigate inequities allowed the agency to seek various methods for displaying and disseminating the data; hire a diverse workforce to perform surveillance, education, and community outreach; improve access to testing and resources; and seek opportunities to lobby and apply for additional pandemic response funding. DCoDPH saw and continues to see the way COVID-19 disproportionately affects communities of color in Durham. As a result, the lessons learned will be integrated into all facets of our work. 

The pandemic has contributed to the health department being more prepared to provide a proactive response to preventing and identifying issues within historically marginalized populations of the community rather than reactive to inequities as they surface. Much of this success has come from working collaboratively and being innovative in data collection, visualization, and dissemination strategies. For example, the innovative process of collecting employment and occupation data led to an expedited response to clusters and outbreaks in vulnerable populations, since the information was available to make those connections. DCoDPH will also expand the use of forward-facing dashboards to provide surveillance for other health conditions, disasters, or crises to allow for greater transparency in the work and instill confidence in community members. The health department developed a COVID-19 public-facing vaccine distribution dashboard to accompany the COVID-19 surveillance dashboard so Durham County residents are aware of vaccinations rates by race, ethnicity, age, gender, geographic location, and vaccine status (partial or full). Another DCoDPH priority is to develop periodic video messages from the health director with evidence-based facts to broaden the scope of information sharing strategies to reach populations with lower literacy levels. 

Advocating for permanent funding is an additional strategy that will allow DCoDPH to build a more robust public health infrastructure long-term to facilitate quick responses to disasters and ensure optimal surveillance capabilities. DCoDPH submitted an Office of Minority Health (OMH) grant to address needs within the county’s most vulnerable populations through education, peer advocacy, and Community Health Workers. The health department is also in the process of designing more appropriate COVID-19 messaging for different ethnic groups and increasing the number of agency partners to secure a readily available workforce of volunteers. In addition, DCoDPH seeks to increase health education surrounding pre-existing conditions, health screenings, and holistic self-care within vulnerable communities so that these communities have the resiliency to combat disease. These initiatives are all contingent upon the fact that DCoDPH can address inequities within the agency and continue to approach systemic racism as a public health issue that is the root cause of the inequities. Only then is the health department equipped to serve the public and mitigate the impact of COVID-19 on vulnerable communities. 

## Figures and Tables

**Figure 1 ijerph-18-06544-f001:**
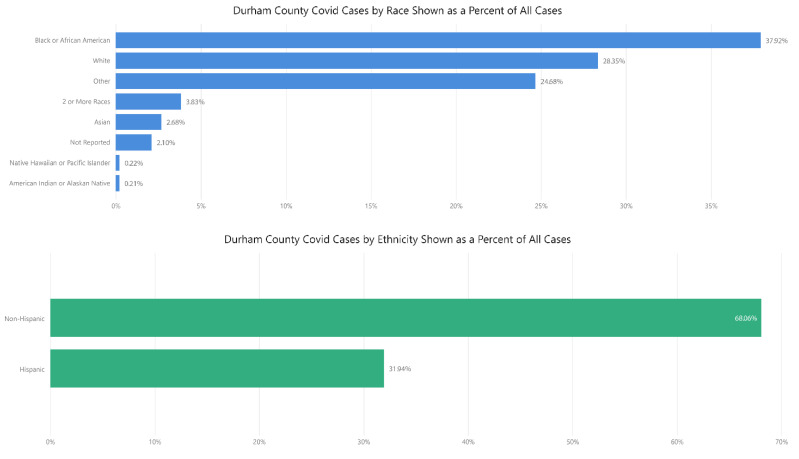
Durham County COVID-19 Confirmed Cases by Race and Ethnicity.

**Figure 2 ijerph-18-06544-f002:**
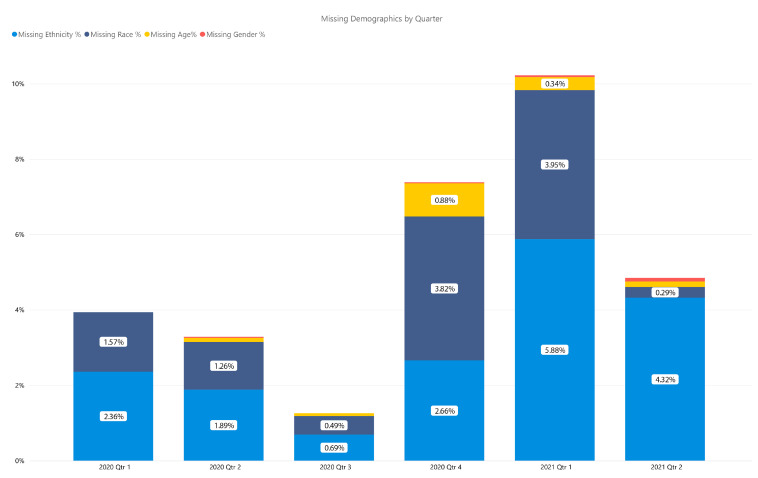
Missing Demographic Information for COVID-19 Cases by Quarter.

**Table 1 ijerph-18-06544-t001:** Durham County Department of Public Health COVID-19 Response Partners.

Partner	Role	Sector
Duke University	Coordinated communication efforts	Academia
Duke Division of Community Health	Coordinated Holton Career and Resource Center testing site	
North Carolina Central University	Coordinated COVID-19 screening, testing, and contact tracing for students, faculty, and staff	
Durham County Emergency Management	Coordinated Durham County COVID-19 emergency response	Local Government
Durham County Department of Social Services	Supported public information, safety, planning, operations, logistics, finance, and administration in the emergency response Incident Command Structure, provided funding for response	
Durham County Cooperative Extension		
Durham County Emergency Services		
Durham County Sheriff’s Department		
Durham County Manger’s Office		
Durham County Detention Facility		
Durham County Attorney’s Office		
Durham County Public Information Office		
Durham County Public Libraries		
Durham County Finance		
Durham County General Services		
Durham County Volunteer Fire Department		
Durham County Criminal Justice Resource Center		
City of Durham Office of Community Development		
City of Durham Fire Department		
City of Durham Police Department		
City of Durham Finance Department		
City of Durham Transportation		
City of Durham Public Information		
Durham Emergency Communications Center		
Bull City United Violence Interruption Program	Conducted community outreach and material delivery	Local Government
North Carolina Department of Health and Human Services	Coordinated testing sites, provided COVID-19 case, surveillance, testing guidance and messaging, provided funding	State Government
Durham Public Schools	Assisted with messaging and coordinated food delivery	Education
Duke University Health System	Provided health services, testing, and accurate COVID-19 information	Healthcare
Lincoln Community Health Center		
Long term care facilities		
Durham Housing Authority	Coordinated testing and community outreach	Housing
El Centro Hispano	Coordinated testing and community outreach, assisted with food and housing resources	Non-profit
American Red Cross	Supported the emergency response Incident Command Structure	
Curamericas	Coordinated and conducted community outreach	
End Hunger Durham	Supported education and resources related to food access	
Food Insight Group		
EAT NC and Durham FEAST		
United Way		
Farmer Foodshare		
Inter Faith Food Shuttle		
Food Bank of Central and Eastern NC		
60+ Food Pantries in Durham County		
Durham Public Schools Student Nutrition Services		
Triangle Nonprofit & Volunteer Leadership Center		
City of Durham GIS Department		
Durham Grab and Go Meals with the Mustard Seed Project		
Go Triangle	Participated in the emergency response Incident Command Structure	Transportation
LATIN-19	Center the community’s voice and coordinate response efforts across partners	Community Collaborative
African American COVID Task Force		
Partnership for Seniors and More		
Homeless service providers	Participate in task force for homeless services	Social Services

**Table 2 ijerph-18-06544-t002:** Visits and Inspections by Establishment Type.

Establishment Type	Total Visits or Inspections	Total Number of Zip Codes Reached	Zip Code with Highest Number of Visits or Inspections (#)
Restaurants	2856	16	27701 (481)
Childcare	220	12	27707 (59)
Tattoo Artists	89	6	27701 (33)
Residential Care	62	8	27707 (13)
Nursing Homes/Hospitals	8	3	27713 (5)
Total	3235	N/A	N/A

**Table 3 ijerph-18-06544-t003:** Demographic Data of Holton Testing Captured in the DCoDPH Line Listing.

Race and Ethnicity (N = 759)	Frequency (%)
Hispanic	378 (49.8)
African American	204 (26.9)
White	54 (7.1)
Unknown	53 (7.0)
Two or More Races	44 (5.8)
Asian	19 (2.5)
Native Hawaiian or Pacific Islander	4 (0.5)
American Indian or Alaskan Native	3 (0.4)

**Table 4 ijerph-18-06544-t004:** Demographic Data of Holton Testing Captured in the DCoDPH Line Listing.

Race	Females		Males		Total	% of Total Population Tested
	Negative	Positive	Negative	Positive		
American Indian or Alaska Native	56	5	58	3	122	0.5%
Asian	409	25	335	15	784	3.4%
African American	3925	274	2856	230	7285	31.3%
Indian	36	1	65	3	105	0.5%
Middle Eastern	36	0	35	1	72	0.3%
Multiracial	262	20	174	9	465	2.0%
Native Hawaiian or Other Pacific Islander	6	0	4	0	10	0.0%
Other	1388	298	1331	293	3310	14.2%
Other Asian	14	1	14	2	31	0.1%
Other Pacific Islander	6	1	3	0	10	0.0%
Unknown	228	28	249	33	538	2.3%
White	5316	198	4826	231	10,571	45.4%
Total	11,682	851	9950	820	23,303	-
**Ethnicity**	**Females**		**Males**		**Total**	**% of Total Population Tested**
	**Negative**	**Positive**	**Negative**	**Positive**		
Not Hispanic	9131	452	7806	422	17,811	79.7%
Hispanic	2050	369	1739	366	4524	20.3%
Total	11,181	821	9545	788	22,335	-

## Data Availability

COVID-19 data on Durham County residents can be found at Public Health Coronavirus Data (arcgis.com). (accessed on 21 April 2021).

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
