# Peer review of "Local Government Approaches to Combating COVID-19 Inequities: A Durham County Department of Public Health Perspective"

_ijerph, 2021, doi:10.3390/ijerph18126544_

Round 1

Reviewer 1 Report

The manuscript describes how a US public health department attempted to manage a COVID-19 pandemic crisis while taking possible differentiating impacts of pre-existing social inequities in the given geographic area seriously.

I would like to congratulate the authors for their exemplarily complex and inclusive intervention approach as well as on writing a timely and clear manuscript about it – as a public health researcher at the time busy with similar COVID-19 issues on a different continent I read the piece with much interest, and I found it both inspiring and instructive.

The only general recommendation I can think of to, perhaps, improve the manuscript even further relates to the only dissonance I felt after reading it: both the title and the objective (89-92) advertise a primary focus on inequity, but the scope of the paper’s presentation of results, especially in the discussion and the conclusions sections is much broader. My feeling is that this issue presents a trade-off dilemma. The formulations of the title and the objective could be broadened (e.g., ‘A collaborative approach to combating COVID-19 across disadvantaged neighbourhoods: …’) to better reflect the paper’s current discussion and conclusions. This would better prepare the reader for all the interesting aspects already discussed in the paper, which do not directly relate to the issues of inequity as such – but would at the same time disallow for adding more details on such specific issues. Alternatively, aspects related directly to the issues of inequity could become elaborated a bit more in the discussion and conclusions sections (see further below) to better match what the current title and objective promise – but that might mean that some other interesting issues now discussed (e.g., staff burnout or data management issues) would have to be omitted to prevent the paper from becoming too lengthy.

Selfishly, based on my own professional interest and focus, I would prefer to see the second option taken. Should the other editors happen to expect the same, I conclude with listing for inspiration also some specific aspects on which I was thrilled to learn more while reading the manuscript, especially its discussion of challenges and the lessons learned (although, some of the aspect could be easily covered also in other parts of the manuscript, including methods):

  • How did you manage personal data protection with respect to ethnic origin / race? (In some countries, it is illegal to create databases from which it is possible to identify any person’s ethnic origin.)
  • How did you go about assessing possible reporting (selection) biases and the validity of individual self-reported epidemiological data across the different social backgrounds / communities?
  • What were your criteria for the identification / ascription of ethnicity / race categories and how did you operationalize them?
  • What precautions did you take to prevent stigmatization of the specific communities upon public dissemination of epidemiological data describing their situation? How effective were these?
  • What specific adjustments did you agree on with your collaborators regarding interventions (e.g., tailoring of the food donation) with respect to the specific communities? Could you give some examples, including the involved justification?

 Specifically in the results section, I would find the presentation as more clearly focused on the issues of inequity with the following adjustment:

  • In the figures (e.g., Fig 1) and tables (e.g., Tab 3), eventual discrepancies between the picked-up infection rates and the actual demographic representations of the specific ethnic groups / communities in the area’s total population could become highlighted, rather than the numbers of cases outside such context

Author Response

Reviewer 1

Comments to the Author

The manuscript describes how a US public health department attempted to manage a COVID-19 pandemic crisis while taking possible differentiating impacts of pre-existing social inequities in the given geographic area seriously.

I would like to congratulate the authors for their exemplarily complex and inclusive intervention approach as well as on writing a timely and clear manuscript about it – as a public health researcher at the time busy with similar COVID-19 issues on a different continent I read the piece with much interest, and I found it both inspiring and instructive.

The only general recommendation I can think of to, perhaps, improve the manuscript even further relates to the only dissonance I felt after reading it: both the title and the objective (89-92) advertise a primary focus on inequity, but the scope of the paper’s presentation of results, especially in the discussion and the conclusions sections is much broader. My feeling is that this issue presents a trade-off dilemma. The formulations of the title and the objective could be broadened (e.g., ‘A collaborative approach to combating COVID-19 across disadvantaged neighbourhoods: …’) to better reflect the paper’s current discussion and conclusions. This would better prepare the reader for all the interesting aspects already discussed in the paper, which do not directly relate to the issues of inequity as such – but would at the same time disallow for adding more details on such specific issues. Alternatively, aspects related directly to the issues of inequity could become elaborated a bit more in the discussion and conclusions sections (see further below) to better match what the current title and objective promise – but that might mean that some other interesting issues now discussed (e.g., staff burnout or data management issues) would have to be omitted to prevent the paper from becoming too lengthy.

Selfishly, based on my own professional interest and focus, I would prefer to see the second option taken. Should the other editors happen to expect the same, I conclude with listing for inspiration also some specific aspects on which I was thrilled to learn more while reading the manuscript, especially its discussion of challenges and the lessons learned (although, some of the aspect could be easily covered also in other parts of the manuscript, including methods):

  • How did you manage personal data protection with respect to ethnic origin / race? (In some countries, it is illegal to create databases from which it is possible to identify any person’s ethnic origin.)

We agreed with the reviewer’s comment. The following explanation was added to 2.2. COVID-19 Data Infrastructure and Case Information:

All COVID-19 case data was managed in compliance with Health Insurance Portability and Accountability (HIPAA) guidelines provided by U.S. DHHS to protect patient confidentiality and protected health information (PHI). All tables that contained any race or ethnicity data were independent of the row level data and did not possess any identifying columns, such as name (first/last), date of birth, and address data. U.S. regulations allow for the use and release of race and ethnicity data for research and educational purposes. The COVID-19 data collected and analyzed falls within those guidelines and was used to make informed detailed decisions. The race and ethnicity information included in the data set allowed DCoDPH to measure trends about populations most impacted in real time, such as increased cases among the Hispanic population, respond quickly, develop tailored responses, and share information with community partners.

  • How did you go about assessing possible reporting (selection) biases and the validity of individual self-reported epidemiological data across the different social backgrounds / communities?

We agreed with the reviewer’s comment. The following explanation was added to 2.2. COVID-19 Data Infrastructure and Case Information:

DCoDPH staff disaggregated the data for various ethnic or racial groups to gather a deeper understanding of trends so biases were not used in decision-making or when developing reports.

  • What were your criteria for the identification / ascription of ethnicity / race categories and how did you operationalize them?

We agreed with the reviewer’s comment. The following explanation was added to 2.2. COVID-19 Data Infrastructure and Case Information:

Race and ethnicity data were self-reported by positive cases. The validity of race and ethnicity self-reporting was verified by those sources providing positive test results and when DCoDPH surveillance staff contacted the positive case. DCoDPH utilized the standard practice for operationalizing race and ethnicity; we communicated with the State of North Carolina Division of Public Health, verified and standardized our methodologies for all case data for reporting purposes. The state of North Carolina also provides our data from a shared database, and we follow their naming conventions on all race and ethnicity data.

  • What precautions did you take to prevent stigmatization of the specific communities upon public dissemination of epidemiological data describing their situation? How effective were these?

We agreed with the reviewer’s comment. The following explanation was added to 2.7. COVID-19 Data Dissemination:

Prior to releasing the dashboard, DCoDPH spoke with community organizations representing the Hispanic and African American populations to identify best methods to display the data and add context to avoid stigmatizing this population. DCoDPH representatives also attended community meetings and met with partners to gain input before the dashboard was published.

  • What specific adjustments did you agree on with your collaborators regarding interventions (e.g., tailoring of the food donation) with respect to the specific communities? Could you give some examples, including the involved justification?

We agreed with the reviewer’s comment. The following explanation was added to 3.4. COVID-19 Resource Allocation:

For example, the Hispanic families typically preferred dry small red beans or black beans while the African American and white families preferred canned pinto or black, so we provided a combination of the two.  The majority of the families wanted whole milk, so that is what was provided.  Most families at the time of food delivery were Hispanic, and this ensured their food preferences were met.    

Specifically in the results section, I would find the presentation as more clearly focused on the issues of inequity with the following adjustment:

In the figures (e.g., Fig 1) and tables (e.g., Tab 3), eventual discrepancies between the picked-up infection rates and the actual demographic representations of the specific ethnic groups / communities in the area’s total population could become highlighted, rather than the numbers of cases outside such context.

We agreed with the reviewer’s comments and perspective. Figure 1 was changed to focus on the race and ethnicity of Durham County COVID-19 cases.

The text above the chart includes the percentage of racial and ethnic groups in relation to the racial group percentage make up of cases. The original Table 3 was deleted. The current Table 3 focuses on demographic data of Holton testing.

Reviewer 2 Report

This paper aims to present local public health department strategies to operationalize and combat COVID-19 inequities, asserting the importance of local government approaches to prevent disease and protect the health of the vulnerable communities. 

The introduction is compelling and the information about the first COVID-19 case being detected in Durham County, North Carolina is insightful.

Line 105- Authors describe the purpose for developing a line listing, though it is not immediately clear what a line listing is.  The authors should first define what a line listing is.  Furthermore, an introductory or transition sentence that establishes the significance of having a line listing will also be helpful.

Based on the title, it seems the premise of the paper is to show how “a local health department operationalized equity in various stages of COVID-19 response” (line 38).  Therefore, the connection between the operationalization of equity and the COVID-19 response needs to be made stronger within the paper.  In the Materials and Methods section where the different approaches are discussed, the authors should describe and reinforce how the approaches presented address health inequities.  Various activities are described, though there should be more direct discussion as to how these activities promote health equity.  In line 82, there is mention of using an equity lens.  That concept needs to be further developed within the paper as to what this entails.  The equity lens concept is mentioned again in line 577, and at that point it should be clearly established what this concept means. 

The ‘COVID-19 Lessons Learned’ section proposes very helpful strategies and seems to expand on the equity lens concept, though it is at the end of the paper.  This concept should also be developed before the end of the paper.  Overall, this paper is a laudable effort that is comprehensive and pertinent.  Attending to some clarifying questions may help to improve the quality of the paper.

Author Response

Reviewer 2

This paper aims to present local public health department strategies to operationalize and combat COVID-19 inequities, asserting the importance of local government approaches to prevent disease and protect the health of the vulnerable communities.

The introduction is compelling and the information about the first COVID-19 case being detected in Durham County, North Carolina is insightful.

Line 105- Authors describe the purpose for developing a line listing, though it is not immediately clear what a line listing is.  The authors should first define what a line listing is.  Furthermore, an introductory or transition sentence that establishes the significance of having a line listing will also be helpful.

We agreed with the reviewer’s comment. An explanation of the line listing is now provided. The following two statements were added to 2.2. COVID-19 Data Infrastructure and Case Information:

The Durham County COVID-19 line listing is an Excel spreadsheet that contains data about each positive case. It is used as a data hub to perform all COVID-19 data analyses and data feed for our public facing dashboards.

Based on the title, it seems the premise of the paper is to show how “a local health department operationalized equity in various stages of COVID-19 response” (line 38).  Therefore, the connection between the operationalization of equity and the COVID-19 response needs to be made stronger within the paper.  In the Materials and Methods section where the different approaches are discussed, the authors should describe and reinforce how the approaches presented address health inequities.  Various activities are described, though there should be more direct discussion as to how these activities promote health equity.  In line 82, there is mention of using an equity lens.  That concept needs to be further developed within the paper as to what this entails.  The equity lens concept is mentioned again in line 577, and at that point it should be clearly established what this concept means.

The ‘COVID-19 Lessons Learned’ section proposes very helpful strategies and seems to expand on the equity lens concept, though it is at the end of the paper.  This concept should also be developed before the end of the paper.  Overall, this paper is a laudable effort that is comprehensive and pertinent.  Attending to some clarifying questions may help to improve the quality of the paper.

We agreed with the reviewer’s comments and statements about the overall focus of the manuscript. The authors made additions to each section to be more explicit about equity throughout the DCoDPH COVID-19 response. Additional details are provided throughout the manuscript including what specific actions were taken and the equity impact.

Round 2

Reviewer 2 Report

The authors effectively addressed reviewer comments to establish a stronger connection between the operationalization of equity and the COVID-19 response within their paper.   With examples and expanded discussion provided by the authors, there is a more distinct emphasis on how the approaches presented address health inequities.  The paper is clearer and appears suitable for publication.